# Giant dynamic electromechanical response via field driven pseudo-ergodicity in nonergodic relaxors

He Qi [1,2], Tengfei Hu[3], Shiqing Deng [2], Hui Liu [2], Zhengqian Fu [3] ✉ & Jun Chen [1,4] ✉

Enhanced electromechanical response can commonly be found during the crossover from normal to relaxor ferroelectric state, making relaxors to be potential candidates for actuators. In this work, $(Pb_{0.917}La_{0.083})(Zr_{0.65}Ti_{0.35})_{0.97925}O_3$ ceramic was taken as a case study, which shows a critical nonergodic state with both double-like *P-E* loop and irreversible relaxor-normal ferroelectric phase after poling at room temperature. The low-hysteresis linear-like $S\text{-}P^2$ loop, in-situ synchrotron X-ray diffraction and transmission electron microscope results suggest that the nonpolar relaxor state acts as a bridge during polarization reorientation process, accompanying which lattice strain contributes to 61.8% of the total strain. In other words, the transformation from normal ferroelectric to nonergodic relaxor state could be triggered by electric field through polarization contraction, which could change to be spontaneously with slightly increasing temperature in the nonergodic relaxor zone. Therefore, pseudo-ergodicity in nonergodic relaxors (i.e. reversible nonergodic-normal ferroelectric phase transition) driven by periodic electric field should be the main mechanism for obtaining large electrostrain close to the nonergodic-ergodic relaxor boundary. This work provides new insights into polarization reorientation process in relaxor ferroelectrics, especially phase instability in nonergodic relaxor zone approaching to freezing temperature.

Relaxor ferroelectrics exhibit excellent electrical properties as well as many controversial problems on structure, thus leading to continuous attention in the past several decades[1–5]. Owing to the different sizes of the compositionally disordered cations and the random electric fields created because of the different charges of these cations in relaxors, a unique feature of polar nanoregions (PNRs) can be detected, giving rise to unique physical properties[6–13]. Even though there are some different models describing the morphology of PNRs[13–18], it is widely accepted that the relaxors can be divided to be ergodic and nonergodic states according to the dynamic of PNRs[10,19–25].

On cooling below the so-called *Burns* temperature ($T_B$), mobile PNRs showing ergodic behavior appears[7,10,20,21]. The term "ergodic" was elaborated from the Greek words by *Boltzmann* in order to describe energy surfaces of dynamical systems in statistical mechanics: the time average is equal to the ensemble average under appropriate conditions[26]. From thermodynamic theory, two different kinds of potential wells of $P = 0$ (nonpolar state) and $P = P1 \neq 0$ (polar state, the

[1]Beijing Advanced Innovation Center for Materials Genome Engineering, Department of Physical Chemistry, University of Science and Technology Beijing, Beijing 100083, P. R. China. [2]School of Mathematics and Physics, University of Science and Technology Beijing, Beijing 100083, P. R. China. [3]State Key Laboratory of High Performance Ceramics and Superfine Microstructures, Shanghai Institute of Ceramics, Chinese Academy of Sciences, Shanghai 200050, P. R. China. [4]Hainan University, Haikou 570228 Hainan Province, China. ✉e-mail: fmail600@mail.sic.ac.cn; junchen@ustb.edu.cn

difference between various long-range ordered ferroelectric states with different polarization orientations is ignored to simplify the analytic process) might exist in perovskites depending on the off-centering of B-site ions in the oxygen octahedron. For the paraelectric state, only $P = 0$ state is stable and $P = P1$ state cannot be reached even under a strong bias field. With decreasing temperature, $P1$ potential wells would become available. Furthermore, the dynamic energy (consisting of thermal activation energy and random field) of the B-site ions is stronger than all the potential energy barriers between these potential wells. At this state, the time average would be the same as the ensemble average because every state can be reached over time, thus exhibiting ergodic behavior. These ergodic PNRs are not stable but create and destruct endlessly in the host lattice[13,27]. As a result, even a long-range ordered ferroelectric state could be triggered under a strong external electric field, the ferroelectric ordering would be disrupted into an initial ergodic state spontaneously after removing the electric field, providing the basic for achieving large electrostrain and excellent energy storage properties[4,7,28–31]. The ergodicity would be broken on further cooling when some of these barriers are so high that the time needed to overcome them is longer than any practically reasonable observation time, thus calling a nonergodic state (including normal ferroelectric phase and nonergodic relaxor ferroelectric phase)[7,13,19]. The nonergodic relaxor state can be distinguished from the long-range ordered ferroelectric state that it makes up with short-range ordered PNRs and shows apparent relaxation in dielectric properties.

It is generally accepted that a ferroelectric state can be induced from a nonergodic relaxor by applying a sufficiently strong electric field to overcome the potential energy barrier accompanying the appearance of an obvious ferroelectric-relaxor transformation peak at $T_{F-R}$ on the dielectric spectroscopy[22,32–34]. A maximum electrostrain can be generated a little above $T_{F-R}$ owing to the spontaneous backward switching from the field-induced ferroelectric state to the initial relaxor state during discharging. The appearance of this reversible phase transformation during electric field cycling suggests that the virgin structure has been transformed into the ergodic relaxor zone. In other words, $T_{F-R}$ should be the same as the critical temperature for the freezing of PNRs from ergodic to nonergodic relaxor state ($T_f$) in typical relaxors, such as Pb(Mg$_{1/3}$Nb$_{2/3}$)O$_3$ and Pb(Zn$_{1/3}$Nb$_{2/3}$)O$_3$, if the reversible relaxor-ferroelectric phase transition is exclusive for ergodic relaxors. However, the investigations in (Bi$_{0.5}$Na$_{0.5}$)TiO$_3$ (BNT), Bi(Mg$_{0.5}$Ti$_{0.5}$)O$_3$ (BMT), and (Pb,La)(Zr,Ti)O$_3$ based relaxors indicate that $T_{F-R}$ is usually smaller than $T_f$[32–34]. Depending on the results of temperature-dependent X-ray diffraction, electrical properties, and second harmonic generation measurements, the transition from field-induced ferroelectric to ergodic relaxor are identified as two separate and discrete processes on heating[32]. For instance, the electrically

induced long-range ferroelectric to ergodic relaxor transition in 0.94BNT-0.06BaTiO$_3$ is continuous over a temperature range of -10 °C starting from 83 to 95 °C[32]. The induced ferroelectric domains first lose their ferroelectric texture and then dissociate into PNRs. By comparing, the temperature range between these two stages is found to be much wider in BMT-based ceramics. For example, the coexistence of nonergodic and ergodic relaxor states is found in a wide temperature range of -150 °C starting from $T_{F-R}$ to $T_f$, causing a temperature insensitive large strain in 0.4BMT-0.3Pb(Mg$_{1/3}$Nb$_{2/3}$)O$_3$–0.3PbTiO$_3$[33]. Yet the model of nonergodic-ergodic relaxors coexistence cannot well explain the critical ferroelectric properties, that the backward switching process of double $P$-$E$ loops need to load a reversed electric field, found in this work. Moreover, the dynamic process of applying the electric field is ignored in this model for identifying the relaxor state.

It seems that the ergodicity can be influenced by the application of an external electric field. Therefore, in this work, a critical composition of (Pb$_{0.917}$La$_{0.083}$)(Zr$_{0.66}$Ti$_{0.34}$)$_{0.97925}$O$_3$ (PL$_{8.3}$Z$_{66}$T$_{34}$) with $T_{F-R}$ slightly above room temperature is studied to reveal the phase structure evolution under electric field in order to reunderstand the origin for achieving high piezoresponse and large electrostrain around nonergodic-ergodic relaxor boundary.

## Results

### Evolution of ergodicity and ferroelectricity on heating

According to the PLZT phase diagram drawn by previous researchers, critical compositions located close to the rhombohedral (R)-tetragonal (T)-relaxor triple point were prepared, as shown in Fig. S1. With increasing La content, a transformation from normal to relaxor ferroelectric state can be detected, along with a drastic increase of piezoelectric response and electrostrain. High $d_{33}$ and electrostrain reach up to 770 pC N$^{-1}$ and 0.27% (@ 4 kV mm$^{-1}$) can be detected in PL$_{8.3}$Z$_{66}$T$_{34}$ and PL$_9$Z$_{67}$T$_{33}$ ceramics, respectively. The previous works usually thought that the achievement of large $d_{33}$ in relaxor ferroelectrics should be ascribed to the local heterostructure, while the large electrostrains should be related to the reversible ergodic relaxor to a normal ferroelectric phase transition. However, the PL$_9$Z$_{67}$T$_{33}$ ceramic shows a high freezing temperature $T_f$ -50 °C, indicating that the studied samples with La content less than 9 mol% cannot be an ergodic relaxor state. Moreover, excellent electrical properties appear at the same time in a narrow composition range with $T_f$ slightly above room temperature; thus, a common mechanism should be existed for achieving large electromechanical properties of both piezoelectricity and strain behavior. In order to get rid of the interference of spontaneously reversible phase transition on the determination of ergodicity, the widely studied PL$_9$Z$_{67}$T$_{33}$ ceramic with a maximum strain and a near-zero d$_{33}$ was not chosen in this work. Instead, a critical composition of PL$_{8.3}$Z$_{66}$T$_{34}$ showing a pinched $P$-$E$ loop, a special electrostrain behavior, and a large $d_{33}$ -770 pC N$^{-1}$ is chosen as a case study.

Typical dielectric relaxation behaviors of both diffuse phase transition and frequency dispersion can be seen around the temperature of dielectric maxima ($T_m$) in this PL$_{8.3}$Z$_{66}$T$_{34}$ relaxor ferroelectric, as shown in Fig. 1. The broken of ergodicity into nonergodicity mainly involves the frozen of PNRs at freezing temperature $T_f$, this critical temperature cannot be seen directly but can be calculated from the Vogel–Fulcher (V–F) law[23]:

$$f = \frac{f_0 \exp E_b}{T_m - T_f}. \tag{1}$$

where $f$ is the measurement frequency, $f_0$ is the Debye frequency, and $E_b$ is the activation energy. According to the fitting results of Curie–Weiss and V–F laws shown in Fig. S2, the ergodic relaxor state exists in the temperature range of 74–305 °C. A dielectric anomaly relating to the transition from field-induced normal ferroelectric phase

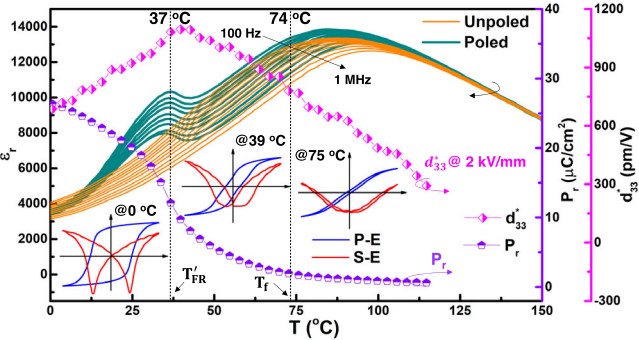

**Fig. 1 | Temperature-dependent on dielectric and ferroelectric properties of PL$_{8.3}$Z$_{66}$T$_{34}$.** The evolution of dielectric permittivity of both unpoled and poled samples, $P_r$, and large electromechanical response $d_{33}^*$ (@ 2 kV mm$^{-1}$) depending on temperature. The insets are the $P$-$E$ and $S$-$E$ curves at various temperatures.

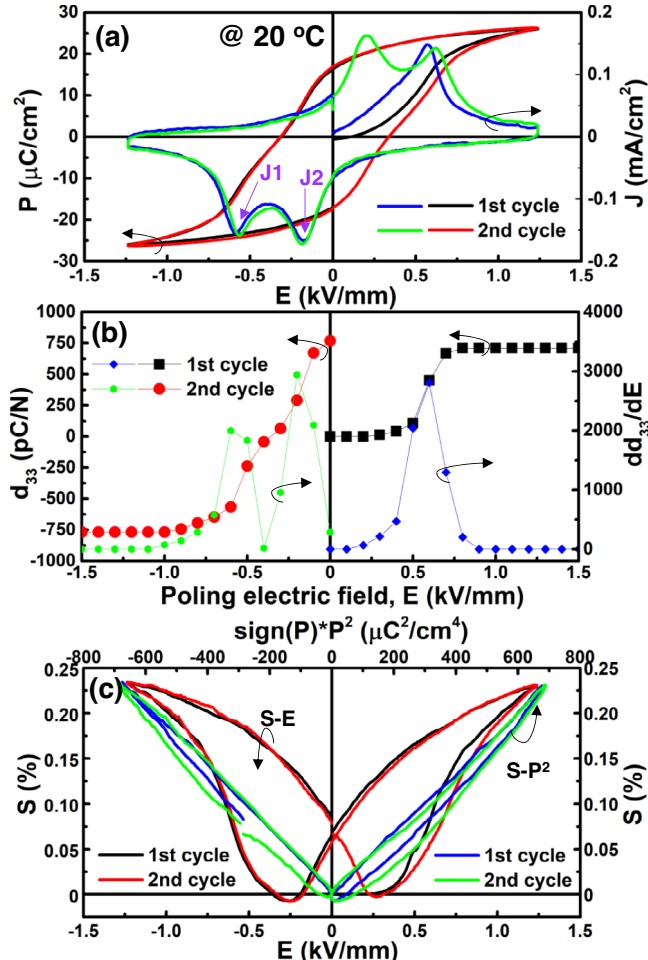

**Fig. 2 | Electrical properties under bipolar electric field. a** Room temperature *P-E* hysteresis loops and associated *J-E* curves, **b** *S-E* curves and associated *S-P²* curves, and **c** $d_{33}$-*E* curves and associated $dd_{33}/dE$-*E* curves measured from first and second electric cycles of PL$_{8.3}$Z$_{66}$T$_{34}$ ceramic. For quasi-static $d_{33}$ measurement, the sample was measured at zero field after being poled by a pulse electric field for 1 s. Especially for the second-cycle $d_{33}$ measurement, the sample was firstly poled by an electric field of 1.5 kV mm$^{-1}$ in the opposite direction.

## Ferroelectric states during electric field cycles

PL$_{8.3}$Z$_{66}$T$_{34}$ ceramic shows a critical state at room temperature, at which both double-like *P-E* loop and irreversible relaxor-normal

ferroelectric phase transformation (large remanent polarization $P_r$ and large $d_{33}$) behaviors can be detected simultaneously, as shown in Fig. 2. This is a universal feature found in different relaxor systems during crossover from nonergodic to ergodic relaxor state[7,34–37]. The position of the *J2* current density peak has been reported to show a time hysteresis effect[30]; thus, the quasi-static piezoelectric coefficient $d_{33}$ was measured after aging for 24 h. A large $d_{33}$ ~770 pC N$^{-1}$ suggests that the electric field-induced long-range ferroelectric ordering is stable after removing the external electric field. Moreover, by applying a reverse electric field on the pooled sample, two sharp $dd_{33}/dE$ peaks can be seen in Fig. 2b, indicating that the polarization reorientation process is no longer domain switching found in normal ferroelectrics. The negligible contribution of domain switching can also be confirmed by a nearly linear *S-P²* response shown in Fig. 2c. The electric field-induced strain, $S_{ij}$, can be expressed as a power series in polarization $P_k$ in tensor notation:

$$S_{ij} = g_{kij}P_k + Q_{ijkl}P_kP_l + \ldots (i,j,k,l) = 1,2,3. \qquad (2)$$

The electric field-induced strain in ferroelectrics originates from polarization rotation and polarization extension/contraction (including the converse piezoelectric effect, the electrostrictive behavior, 180°-domain switching, and electric induced nonpolar-polar phase transition). However, polarization rotation behavior would lead to the large hysteresis of the *S-P²* response, while the converse piezoelectric effect would bring about the deviation from the linear *S-P²* response. According to nearly linear *S-P²* with slight hysteresis, nonpolar-polar phase transition should be the most likely mechanism for the polarization reorientation process in this critical sample.

## Evolution of PNRs under electric field

To check this hypothesis, PNRs' structure, as well as their evolution process under an electric field, were analyzed by using atomic-resolution scanning transmission electron microscopy (STEM) and in situ TEM, as shown in Figs. 3 and 4, respectively. According to the high-resolution TEM images in Fig. 3a and S3, frozen PNRs showing short-rod shape with widths of ~10–25 nm can be detected in different zones[6,7,25]. Moreover, the slush-like PNRs without obvious pseudocubic (PC) matrix can be detected[6,10], leading to the striped distribution of these nonergodic PNRs instead of long-range ordered ferroelectric domains or "hopeless messes" ergodic state. In order to clearly reveal the local structure information of PNRs, atomic-resolution high-angle annular dark-field (HAADF) STEM was taken and quantitatively analyzed, as shown in Fig. 3b, c for the polarization mapping results of different regions in the same grain. Owing to the relatively small field of view for each HAADF STEM image, it is not suitable to extract the size information of PNRs. For example, R symmetry occupies the most regions in Fig. 3b, while the coexistence of R and T symmetry PNRs with preferred orientation distribution can be seen in Fig. 3c. In order to show the local symmetry more clearly, the polarization displacement vectors of 15 different regions were extracted and shown in Fig. 3d. On the one hand, the nonpolar regions found in Fig. 3c only occupies a very small proportion in the grain; thus they should be related to polarizations out of the plane instead of PC phases. On the other hand, R and T symmetries, as well as their medium monoclinic phase, can be seen, which is complied with the design objectives for large electromechanical response at the morphotropic phase boundary in Fig. S1. Therefore, the local structure of this PL$_{8.3}$Z$_{66}$T$_{34}$ relaxor coincides with the "slush-like polar structrues" model: multi-domain structure coexisting with a high density of low-angle domain walls[10]. In addition, *(ooo)/2*-type superlattice spots can be seen along <112>$_C$ direction in Fig. 4, indicating that the local R phase should be the *R3c* space group. After applying an electric field, nearly a single domain is formed in the studied grain, relating to the formation of a long-range

to relaxor state appears at $T'_{F-R}$ ~37 °C. A similar difference between $T'_{F-R}$ and $T_f$ has also been found by other researchers in various relaxors ferroelectric systems[32–34]. Interestingly, the *P-E* loop gradually changes from a square shape to a pinched double-like shape and then to a slim shape on heating in the nonergodic relaxor phase zone. A drastic decrease of remanent polarization ($P_r$) can be detected approaching $T'_{F-R}$, accompanying which the positive strain ($S_{pos}$) increases monotonously upon heating. The $S_{pos}$ reaches its maximum at ~42 °C ($d_{33}^* > 1100$ pm V$^{-1}$ @ 2 kV mm$^{-1}$), at which the negative strain disappears and the electric field-induced relaxor to normal ferroelectric phase transformation could be totally reversible. Yet the electric field-induced reversible relaxor-normal ferroelectric phase transition is widely thought to be the proprietary property of ergodic relaxors; thus, the state above $T'_{F-R}$ used to be marked as an ergodic relaxor state sometimes. In order to eliminate the possibility that there is an error in determining $T_f$ value, a state slightly below $T'_{F-R}$ (namely, the room temperature state of PL$_{8.3}$Z$_{66}$T$_{34}$ ceramic) is taken for further analysis.

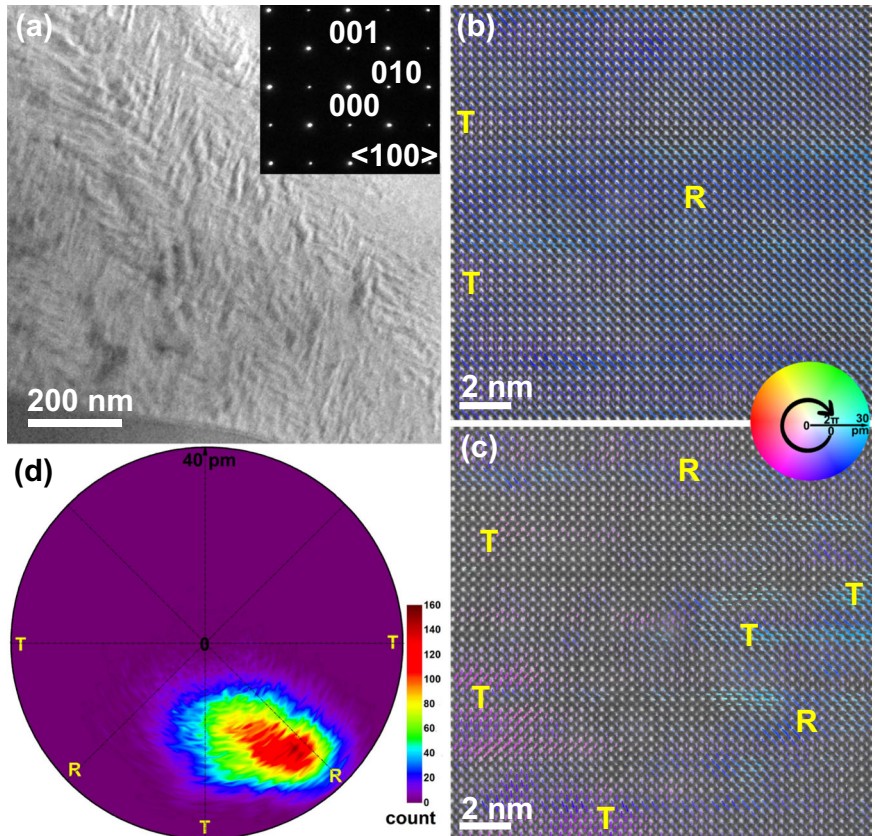

**Fig. 3 | Local structure information of PNRs. a** Bright-field TEM image and SAED pattern and **b**, **c** atomic-resolution HAADF STEM polarization mappings measured at different regions along <100>$_c$ direction for PL$_{8.3}$Z$_{66}$T$_{34}$ ceramic. **d** The statistics of polarization displacements extracted from 15 HAADF STEM images (including more than 60,000 atoms) measured at different regions in a single grain. The colors display the density distribution.

ferroelectric ordering state. In addition, the intensity of (ooo)/2-type superlattice spots becomes weakened and nearly disappeared under 40 V. After removing the external electric field, most of the ferroelectric ordering can be maintained, as shown in both bright and dark field images. At the same time, a small amount of PNRs with weak (ooo)/2 superlattice spots can be observed. During further loading electric field along the inverse direction, decreased domain size, as well as increased intensity of (ooo)/2 superlattice spots, can be detected before the formation of a long-range ferroelectric state again, validating the increase of the amount of PNRs found from bright-field images. Namely, the PNRs state is an intermediate bridge during the process of domain reorientation during applying a reverse electric field.

In situ high-energy synchrotron X-ray diffraction was carried out to further discover this phenomenon at a macro scale. The structure change under an electric field can be identified by the in situ SXRD patterns extracted from the 45° sector, where the influence of the texture can be negligible[38], as shown in Fig. 5a for representative (222)$_{pc}$ and (400)$_{pc}$ diffraction peaks and in Fig. S4 and Table S1 for several typical Rietveld refinement results with the highest reliability among the use of different space groups. Single (222)$_{pc}$ and (004)$_{pc}$ diffraction peaks at different azimuthal angles can be seen in the virgin sample, this is a common feature in relaxors that the PNRs are too small to be resolved by XRD. Even though each PNR is undergoing unique distortions at a local scale (Fig. 3b) and applying a single symmetry is not ideal for revealing the real local system, but it is still the best option to identify this initial relaxor state as a PC phase for average structure analysis[39,40]. With applying an electric field over 0.6 kV mm⁻¹, apparent movement to a lower angle and widening of diffraction peaks can be measured. Because each grain orientation

must undergo separate and unique distortions to the applied field, the 45° sector will inevitably look like a low symmetry structure due to the convolution of these effects. However, in order to simplify the analysis process and provide the basis for quantitative analysis of strain contribution, the best refinement result using the monoclinic M (*Cm* space group) phase is selected here to identify the high-field ferroelectric state. The ferroelectric M phase can be maintained till removing the external electric field, further confirming the nonergodicity of the studied sample at room temperature. The diffraction peaks gradually become narrow and shift to a higher angle when a weak negative electric field is loaded, indicating the backward switching to the initial nonpolar nonergodic relaxor state. An M ferroelectric phase can be triggered with a further increasing electric field, as indicated in Fig. S4. The evolution of phase fraction with changing electric field is shown in Fig. 5c. It can be found that the nonpolar PC phase acts as a medium for polarization reorientation. Moreover, this polarization reorientation process should be driven by a negative electric field instead of spontaneously. Together with the patterns extracted from the 90° sector revealed in Fig. S5, little domain switching behavior can be observed, being consistent with the nearly linear $S$-$P^2$ response.

## Discussion

In order to understand the contributions to the large electromechanical response close to the studied critical composition, quantification of lattice strain $S_{latt}$, volumetric strain $S_{V,33}$ caused by the phase transformation and domain texture strain $S_{text,33}$ owing to the texture of the M domains along electric field direction was taken according to the relative change of diffraction peak intensity and position at different azimuthal angles shown in Figs. S6, S7[38-41]. It is interesting that the M phase exhibits a large change in multiples of a

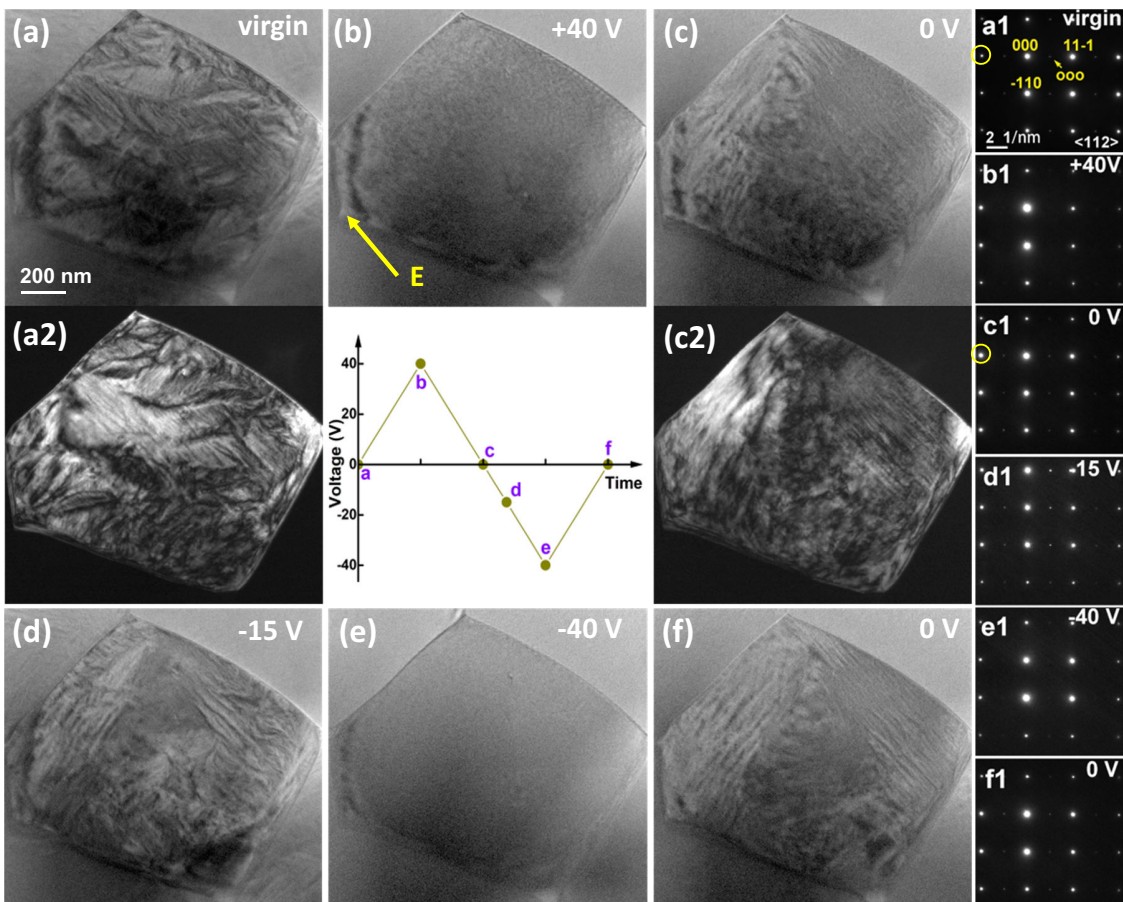

**Fig. 4 | Domain morphology of the PL$_{8.3}$Z$_{66}$T$_{34}$ ceramic under different electric field conditions at room temperature. a** virgin, **b** 40 V, **c** 0 V, **d** −15 V, **e** −40 V, **f** 0 V, and **a1**–**f1** corresponding SAED. Dard field images for the **a2** virgin and **c2** poled states by using the circled diffraction spots in **a1** and **c1**, respectively.

random distribution $\Delta f_{hkl}$ for (400)$_C$ at $\phi = 0°$ (Fig. S7) when it starts to be transformed from the PC phase at electric field ~0.4 kV/mm. This result suggests that the PC phase directly transforms into a highly textured M state through polarization extension instead of a randomly distributed polarization state, thus little macrodomain rotation can be decomposed from $S$-$P^2$ loops in Fig. 2c and synchrotron XRD in Fig. S5. According to the calculated strain results under different electric fields in Figs. 5d, f, the total calculated strain $S_{cal}$ (=$S_{latt} + S_{V,33} + S_{text,33}$) and the detected strain $S_{det}$ are in good agreement, and 61.8, 22.8, and 15.4% of the total macroscopic strain comes from $S_{latt}$, $S_{V,33}$ and $S_{text,33}$, respectively. This is quite different from the quantification results of BNT-based phase-transition relaxor ferroelectrics, in which the large lattice distortion degree (0.342% for the R phase and 1.263% for the T phase) of the field-induced ferroelectric phases contributes to large domain switching and phase transition strains but the poor piezoelectricity ($d_{33} < 200$ pC/N) only contributes 32% lattice strain of the total strain[40]. The much smaller lattice distortion of the PLZT system leads to a much smaller total strain as well as a much lower driven field and strain hysteresis than that of BNT-based ceramics. Together with a large lattice strain response, a large piezoelectric response and low field-driven large strains can be achieved in the studied PLZT compositions.

Ignore the impact of anisotropy between different ferroelectric phases, the Landau free energy along electric field direction for a stress-free ferroelectric can be expressed as an expansion in terms of polarization, terminated here at terms of the sixth power[42]:

$$F = \alpha P^2 + \beta P^4 + \gamma P^6. \tag{3}$$

Thus, the $P$-$E$ loop can be fitted by

$$E = 2\alpha P + 4\beta P^3 + 6\gamma P^5. \tag{4}$$

According to the achieved parameters, the equation of Landau–Devonshire theory in three dimensions can be given as follows:

$$F\left(\vec{P}\right) = F_0 + a\left(P_x^2 + P_y^2 + P_z^2\right) + b_1\left(P_x^4 + P_y^4 + P_z^4\right) + b_2\left(P_x^2 P_y^2 + P_x^2 P_z^2 + P_y^2 P_z^2\right)$$
$$+ c_1\left(P_x^6 + P_y^6 + P_z^6\right) + c_2\left[P_x^4\left(P_y^2 + P_z^2\right) + P_y^4\left(P_x^2 + P_z^2\right) + P_z^4\left(P_x^2 + P_y^2\right)\right] + c_3 P_x^2 P_y^2 P_z^2. \tag{5}$$

In order to explain the evolution of $P$-$E$ shape, especially for the double $P$-$E$ loop around Curie temperature of BaTiO$_3$ with changing temperature, the phenomenological theory of thermodynamics was proposed to fit the $P$-$E$ loops using the Landau–Ginzburg–Devonshire theory[43]. In this work, the free energy profile of a ceramic with a special double-like $P$-$E$ loop is fitted based on the reference of previous research results and parameters[44], in order to qualitatively analyze the relative free energy state between relaxor and normal ferroelectric phases. Figure 6a–c exhibit the fitted $P$-$E$ loop as well as free energy profiles calculated by using phase-field simulation of PL$_{8.3}$Z$_{66}$T$_{34}$ ceramic at room temperature. It is clear that the thermal activation energy cannot overcome the barrier between the nonpolar and polar state, leading to the nature of nonergodicity. The large energy barrier at a nonpolar ($P = 0$) state in normal ferroelectrics leads to quick polarization rotation around the coercive electric field. Differently, deep free energy well relating to nonergodic relaxor state with

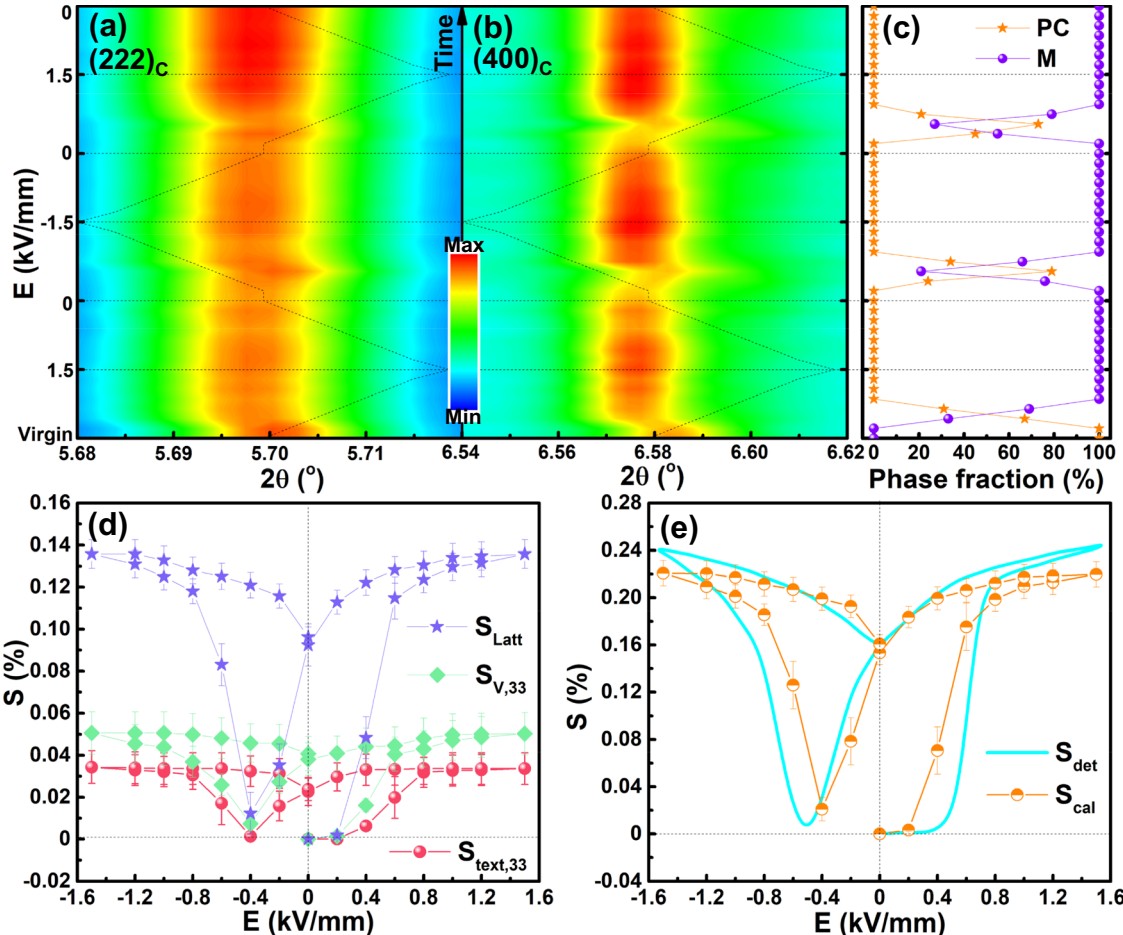

**Fig. 5 | Quantification of structure evolution under electric field.** Evolution of **a** $(222)_C$, **b** $(400)_C$ diffraction lines and **c** phase fraction for $PL_{8.3}Z_{66}T_{34}$ sample under different external electric fields. Quantification of **d** lattice strain $S_{latt}$, volumetric strain $S_{V,33}$, domain texture strain $S_{text,33}$ and **e** total calculated strain $S_{cal}$ and detected strain loop $S_{det}$. The data are presented as the mean ± SEM.

macroscopic nonpolar would be formed at around $T'_{F-R}$, acting as a bridge during polarization reorientation, as can be seen clearly from the diagram of polarization evolution in Fig. 6d. Taking the electric field cycle as the statistical time, two polar states along electric field direction with opposite polarization vectors and nonpolar state could be reached, thus it can be called as pseudo-ergodicity. This further indicates that the reversible transformation between relaxor and normal ferroelectric could also be realized in nonergodic relaxors. Moreover, the backward switching from high-field long-range ordered ferroelectric phase to the initial nonergodic relaxor state changes to be spontaneously upon heating above $T'_{F-R}$, being the basic for generating large electrostrain behavior. These features are quite different from that found in normal ferroelectrics.

In conclusion, the $PL_{8.3}Z_{66}T_{34}$ ceramic with ergodic relaxor zone in 74–305 °C was taken as a case to reunderstand the origin of large electrostrain accompanied by electric field-induced relaxor-normal ferroelectric phase transition. The stability of the ferroelectric phase in the poled state decreases on heating in nonergodic relaxor zone, especially, a unique state that a double-like $P$-$E$ loop and a large quasistatic $d_{33}$ appear simultaneously at room temperature. Linear and low-hysteresis $S$-$P^2$ response indicates that little polarization rotation contributes to the polarization reorientation process. Together with in situ high-energy synchrotron XRD and TEM results, the electric field triggered polarization extension/contraction by using an initial nonergodic relaxor state as a medium plays the main role in the polarization reorientation. The results further suggest that the sharply decreased remanent polarization and negative strain at around $T_{FR}'$ should be originated from a reversible phase transition between nonergodic relaxor and long-range ordered ferroelectric phase, which is no longer the private characteristic for ergodic relaxors. The newly discovered pseudo-ergodicity under periodic electric fields in nonergodic relaxors would provide a better understanding for designing high-performance relaxor ferroelectrics.

## Methods

### Sample preparation

A conventional mixed oxide route was utilized to prepare $Pb_{1-3x/2}La_x(Zr_{1-y}Ti_y)_{1-x/4}O_3$ ($PL_{100x}Z_{100-100y}T_{100y}$) ceramics. High-purity oxides or carbonates, PbO (99.9% in purity), $TiO_2$ (99.9%), $ZrO_2$ (99.9%), and $La_2O_3$ (99.9%), were used as starting raw materials. All powders were weighed and put in a nylon jar together with anhydrous ethanol as the medium, following a planetary ball milling process for 12 h. The dried powder mixtures were calcined at 850 °C for 4 h. The powders were mixed for 24 h again in a planetary ball mill to ensure a good distribution of the dopant, and to reduce the particle size for better densification as well. The sample pellets compacted under 100 MPa were sintered at 1200 °C in the air for 2 h. Then silver paste was painted on both major sides of the sintered pellets and then fired at 550 °C for 30 min.

### Electrical properties

An LCR meter (E4980A, Agilent, Santa Clara, CA) was used to measure the dielectric properties as a function of temperature and frequency.

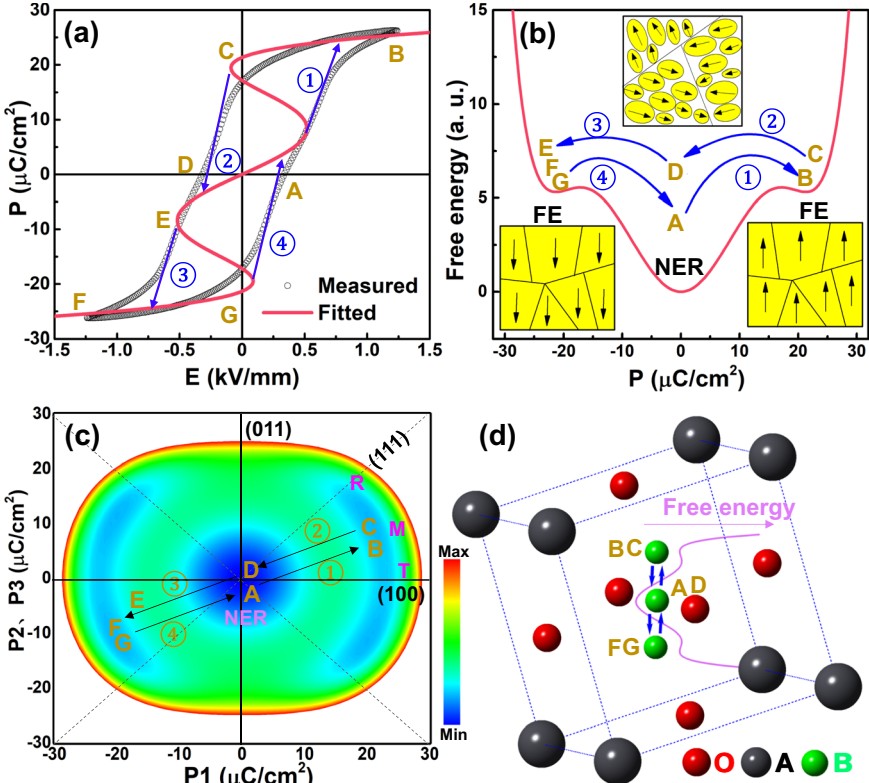

**Fig. 6 | Diagram of electric field-induced pseudo-ergodicity feature. a** The measured and fitted *P-E* hysteresis loops at room temperature. **b**, **c** The Landau free energy profiles and the evolution of the schematic representation of domains in the PL$_{8.3}$Z$_{66}$T$_{34}$ ceramic. **d** Sketch map for the electric field-induced polarization reorientation process through extension/contraction using relaxor ferroelectric state as a bridge in this nonergodic relaxor PL$_{8.3}$Z$_{66}$T$_{34}$ ceramic.

The ferroelectric measuring system (aixACCT, TF Analyzer 1000, Aachen, Germany) was used to measure bipolar polarization vs. the electric field (*P-E*) hysteresis loops and strain *vs.* the electric field (*S-E*) curves at different temperatures. A quasi-static $d_{33}$ meter was used to measure the piezoelectric coefficient $d_{33}$ (China Academy of Acoustics, ZJ-3).

## TEM

The in situ domain morphology observation and selected area electron diffraction (SAED) under varying electric fields were carried out on a field-emission transmission electron microscope (FE-TEM, JEM-2100F, JEOL, Japan) operated at 200 kV with a home-developed electric-holder. Specimens for TEM measurement were prepared by a conventional approach combining mechanical thinning and, finally, Ar$^+$ ion-milling in a Gatan PIPS II.

## STEM

The atomic-scale HAADF STEM images were carried out on an atomic-resolution STEM (aberration-corrected Titan Themis G2 microscope) using a semi-convergence angle of 17 mrad and collection angles of 48–200 mrad. The images were acquired under conditions of fast scanning and cross-correlation summing of multiple frames to minimize sample drift. In order to get the polarization displacement of each atom, the accurate atomic column positions in the STEM images were first precisely extracted by using 2D Gaussian fitting. Then the polarization displacement of each atom can be calculated from the relative displacements between the actual position and the ideal center of the neighboring atoms at hetero sites[45]. Namely, the net displacements of the Pb/La (Zr/Ti) sublattice were extracted with respect to their nearest Zr/Ti (Pb/La) sublattice. The polarization mapping was calculated by customized MATLAB scripts. The microscopy results can be easily reproduced by using the abovementioned methods.

## In situ synchrotron XRD

In situ X-ray measurements were taken at beamline 11-ID-C of APS with high-energy X-ray radiation ($\lambda = 0.1173$ Å) in forwarding scattering geometry on a PerkinElmer amorphous silicon 2D detector, as shown in Fig. S8. Rietveld refinements were performed by using the program Fullprof.

## Data availability

The data that support the findings of this study are available from the corresponding author upon reasonable request.

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

## Acknowledgements

H.Q. and T.H. contributed equally to this work. This work was supported by the National Key R&D Program of China (Grant No. 2022YFB3204000 (J.C.)), the National Natural Science Foundation of China (Grant Nos. 52172181 (H.Q.), 22161142002 (J.C.), 21825102 (J.C.), and 22105017 (H.Q.)), China Postdoctoral Science Foundation (Grant Nos. 2020M680345 (H.Q.) and 2021T140048 (H.Q.)), and the Fundamental Research Funds for the Central Universities, China (Grant No. 06600078 (H.Q.)).

## Author contributions

H.Q. and T.H. fabricated the samples and tested their electrical properties. S.D. and Z.F. did the TEM experiments. H.L. performed SXRD characterizations. The manuscript was drafted by H.Q. and revised by J.C. The work was conceived and designed by J.C. All authors contributed to the discussions and paper preparation.

## Competing interests

The authors declare no competing interests.
