## [Peer Review File · Nature Communications]

Giant dynamic electromechanical response via field driven pseudo-ergodicity in nonergodic relaxorsREVIEWER COMMENTS

Reviewer #1 (Remarks to the Author):

Understanding the structure-property correlations is critical for relaxors, especially to explain the origin of their large electromechanical response. Several reports can be found in literature on this topic, attempting to connect the domain structures and their dynamics with properties. In this study, Qi et. al explains the origin of large electromechanical response in nonergodic relaxor ferroelectric PLZT ceramics. The ergodicity in these materials has been found to be influenced by electric field and temperature previously in literature. Large electromechanical response has been also reported near the crossover from the nonergodic to ergodic state. Here, using a combination of techniques such as in situ synchrotron X-ray diffraction, transmission electron microscopy, P-E loop, dielectric and S-E measurements, the domain structure and their dynamics has been revealed to explain the electromechanical response in nonergodic relaxor state. The use of several techniques together is a great idea to explain the complex correlations in relaxors. Even this paper presents an extensive amount of information from different techniques, several interpretations of the experimental data are still unclear. Also, where this work stands in terms of previous literature is also missing in the current draft. I would suggest a major revision for the current draft before publication.

1. It is unclear why PL8.3Z66T34 composition has been selected for extensive characterization in this study over PL9Z67T33?

2. Currently, Figure 1 has too much information in one plot and it is very hard to read different details. I would suggest breaking up the figure 1 in different subplots.

3. How much d_{33} value changes across La composition especially PL8.3Z66T34 Vs PL9Z67T33? I can only find the value for PL8.3Z66T34 in the current draft.

4. Interpretation about PNRs from the bright-field image in Figure S2 is not clear. How the ellipsoid shape has been obtained for the PNRs? Figure S2, shows different contrast across the image, what is the reason for the differences in the top and bottom regions of the image? How much diffraction contrast from sample misaligns influence the domain interpretation from bright-field images.

5. A high-resolution STEM imaging can be helpful here to understand the polarization changes and determine the domain shapes instead of just bright-field image.

6. Figure 3e label should be -40 kV rather than 0V?

7. Which diffraction spots have been used to obtain the dark field images in Figure 3 a2 and c2?

8. From the in situ high-energy synchrotron X-ray diffraction study, pseudocubic phase has been identified at the zero electric field which transforms to monoclinic phase at higher field. The phase diagram in Figure S1a revealed the composition used in this study are from the boundary between rhombohedral and tetragonal phases. It is unclear if there is any signature of rhombohedral or tetragonal phases in the diffraction data at zero field sample.

9. The fitting of P-E loop in Figure 4d is unclear. Please explain the meaning of the fitted P-E loop in the figure.

10. Explain where the findings in this study stand with previous phase field modeling work by Li, F. et al. The origin of ultrahigh piezoelectricity in relaxor-ferroelectric solid solution crystals. Nat. Commun. 7, 13807 .

Reviewer #2 (Remarks to the Author):

The paper reports results of electric, TEM, and XRD measurements of a nonergodic relaxor (Pb_{0.917}La_{0.083})(Zr_{0.65}Ti_{0.35})_{0.97925}O₃ ceramic sample to demonstrate that the electric field induced relaxor-normal ferroelectric phase transition is the origin of its large electrostrain. The double-like P-E loop and changes of TEM images and XRD patterns under electric field clearly show the relaxor-normal ferroelectric phase transition. The pseudo-ergodicity produced by the field induced phase transition in the nonergodic relaxor is informative to understand electromechanical responses of relaxor ferroelectrics. However, the field induced relaxor-normal ferroelectric phase transition is not surprising and therefore lacks the novelty for publication in Nature Communication. In addition, the following points should be addressed.

T_f in the V-F law of eq. (1) is used without its definition in the introduction. The definition of T_f should be added in the introduction.

T in eq. (1) does not match with T_m in the inset of Fig. 1. It should be replaced by T_m.

E in the line 122 does not match with E_b in eq. (1). It should be replaced by E_b.

The authors analyzed the XRD patterns under electric fields using Rietveld refinements. However, powder diffraction data under an electric field cannot be usually analyzed by single phase Rietveld refinements. No refined parameters and reliable factors of the analyses are shown in the manuscript. The authors should give the detail of the methods and results of the Rietveld refinements.

Reviewer #3 (Remarks to the Author):

The manuscript presents extensive structural, microstructural, electrical and electro-mechanical property data from a PLZT relaxor composition. The main claim of the manuscript, as far as I understand, is that the Authors report a material that is nominally in the non-ergodic relaxor state according to electrical characterisation, however it experiences a reversible electric field induced strain mechanism that is not consistent with past observations in this material state. Overall, I would say the data are reasonable and likely support the conclusions. However, I would also suggest significant improvement could be made in the manuscripts clarity by at least addressing the queries below.

I don't clearly see the differentiation between the strain mechanism shown here and those in other systems. Electric field induced lattice strains (without clear phase transitions) are observed in compositions of NBT-BT-KNN and NBT-ST for example. Applying a similar analysis (of the 45 degree azimuth from HEXRD) in those cases would likely yield a similar result. Can the Authors be more clear about what is unique here?

The Authors use the description "canonical relaxors" with no explanation of their meaning of this description. Please ensure its provided on first use of the word.

The Authors reference the "...nonergodic and ergodic relaxor states coexist in a wider temperature range..." at the bottom of page 3. However, this doesn't seem to match their explanation of what these states are in the previous sections. How do the Authors reconcile that result with the description of these relaxor states they've provided? Perhaps this is the first sentence of the first paragraph on page 4 where they state, "It seems that the ergodicity can be influenced by the application of external electric field.", but please expand on this to explain what is meant?

Page 5 the Authors refer to Figure 1b. However the figure has no b). I suspect this is just referring to the inset.

The Authors state several times that the response is not consistent with polarisation or converse piezoelectric effect, however then go on to use the words such as "...new insights into the polarization reorientation process...". What does reorientation mean if not rotation or intrinsic converse behaviour, this is not at all clear. At the very least a schematic diagram of what the Authors suggest would be required, how are they differentiating these mechanisms from others?

Presumably looking at the XRD data in sector 0 of Fig S5 would provide some insight as to the field induced process. These data are shown in Figure S4, but without any analysis on peak intensities or widths with field. A plot of each of the peak widths vs field would provide some information about particular distortions occurring, if they indeed are.

The analysis of the 45 degree XRD sector to conduct phase analysis is a little odd. Particularly in this state where each grain orientation must be undergoing separate and unique distortions to the applied field, the 45 degree sector will inevitably look like a low symmetry structure due to the convolution of these effects. Saying it is monoclinic here is possibly misleading. The authors should clearly state that each grain orientation set is undergoing unique distortions and applying a single symmetry is not ideal, but perhaps is the best option without employing a more complete analysis method such as using the strain orientation distribution functions as done by other Authors.

The Authors should not use non-scientific language such as "marvelous" to describe the materials behaviour/state.

Related, the English quality of the paper could be improved significantly. There are many grammatical and typographical errors throughout. A quality technical English language edit would be highly advised before resubmitting.

Dear Editor:

Thank you very much for giving us an opportunity to revise our manuscript entitled “**Giant dynamic electromechanical response via field driven pseudo-ergodicity in nonergodic relaxors (Ms. No. NCOMMS-22-44609)**”. We also thank the reviewers for their constructive and valuable comments. We have addressed the comments raised by the reviewers, and the revisions were highlighted in red in the revised manuscript. Point by point responses to the reviewers’ comments are also listed below.

Reviewer #1 (Remarks to the Author):

Understanding the structure-property correlations is critical for relaxors, especially to explain the origin of their large electromechanical response. Several reports can be found in literature on this topic, attempting to connect the domain structures and their dynamics with properties. In this study, Qi et. al explains the origin of large electromechanical response in nonergodic relaxor ferroelectric PLZT ceramics. The ergodicity in these materials has been found to be influenced by electric field and temperature previously in literature. Large electromechanical response has been also reported near the crossover from the nonergodic to ergodic state. Here, using a combination of techniques such as in situ synchrotron X-ray diffraction, transmission electron microscopy, P-E loop, dielectric and S-E measurements, the domain structure and their dynamics has been revealed to explain the electromechanical response in nonergodic relaxor state. The use of several techniques together is a great idea to explain the complex correlations in relaxors. Even this paper presents an extensive amount of information from different techniques, several interpretations of the experimental data are still unclear. Also, where this work stands in terms of previous literature is also missing in the current draft. I would suggest a major revision for the current draft before publication.

1. It is unclear why $PL_{8.3}Z_{66}T_{34}$ composition has been selected for extensive characterization in this study over $PL_9Z_{67}T_{33}$?

Reply: The $PL_{8.3}Z_{66}T_{34}$ composition was specially selected in this work owing to its critical state: **nonergodic relaxor state + field triggered reversible phase transition**. PLZT is a famous system with excellent electrical properties, especially for the large electrostrain under a low electric field in $PL_9Z_{67}T_{33}$ ceramic (Fig. S1). In order to clarify the origin of large strain in relaxor ferroelectrics,

many researches have been done based on the compositions with maximum strain (sprout shape S-E curve with zero negative strain), such as $\text{PL}_9\text{Z}_{67}\text{T}_{33}$ and $0.92\text{BNT}-0.06\text{BT}-0.02\text{KNN}$. Owing to the spontaneously backward switching from high-field ferroelectric state to initial state, these compositions were usually considered as ergodic relaxor state, namely the large strains come from reversible ergodic relaxor-normal ferroelectric phase transition. However, there is a contradiction that these compositions with maximum strain in each system can be identified as nonergodic relaxor state below T_f by using V-F law. It is difficult to distinguish directly between ergodic and nonergodic relaxor states by using other structure and properties analysis methods, thus many researchers consider that this contradiction might be the error of V-F law or even unscientific method of V-F law. **Therefore, if $\text{PL}_9\text{Z}_{67}\text{T}_{33}$ was chosen as the research target, it is hard to give strong proof of reversible nonergodic-normal phase transition.** In this work, there is no doubt that the **$\text{PL}_{8.3}\text{Z}_{66}\text{T}_{34}$ composition is a nonergodic relaxor at room temperature** (not only $T_f \sim 74^\circ\text{C}$ but also large $d_{33} \sim 770$ pC/N and large $P_r \sim 20$ $\mu\text{C}/\text{cm}^2$), **the pinched P - E loop and in-situ structure analysis results suggest reversible phase transition under electric field for this nonergodic relaxor state.** Based on this new finding that nonergodic relaxor state can also reversibly transform into normal ferroelectric state under a periodicity electric field, **we can further confirm that the achievement of large electrostrain, such as $\text{PL}_9\text{Z}_{67}\text{T}_{33}$, should be ascribed to the pseudo-ergodicity in nonergodic relaxor state instead of ergodic-normal ferroelectric phase transition.** A few discussions about the chosen of $\text{PL}_{8.3}\text{Z}_{66}\text{T}_{34}$ composition have been added in text **in red (Pages 4-5)**. Thanks for the comment.

2. Currently, Figure 1 has too much information in one plot and it is very hard to read different details. I would suggest breaking up the figure 1 in different subplots.

Reply: Yes, thanks for the good suggestion, Fig. 1 has been redrawn for easier understanding.

3. How much d_{33} value changes across La composition especially $\text{PL}_{8.3}\text{Z}_{66}\text{T}_{34}$ Vs $\text{PL}_9\text{Z}_{67}\text{T}_{33}$? I can only find the value for $\text{PL}_{8.3}\text{Z}_{66}\text{T}_{34}$ in the current draft.

Reply: The evolution of d_{33} with changing La content can be seen in the following figure. Owing to the spontaneously transformation from high-field ferroelectric to relaxor state during discharging, $\text{PL}_9\text{Z}_{67}\text{T}_{33}$ exhibits a large strain and a weak piezoelectric response of $d_{33} \sim 10$ pC/N. A maximum

$d_{33} \sim 830$ pC/N can be detected in $PL_8Z_{65}T_{35}$. Between $PL_9Z_{67}T_{33}$ and $PL_8Z_{65}T_{35}$, the studied $PL_{8.3}Z_{66}T_{34}$ composition shows not only large piezoelectric response but also field driven reversible phase transition, this critical phenomenon was studied in this work. The results suggest that both large electrostrain and piezoelectric response close to the nonergodic-ergodic relaxor phase boundary should be related to the field induced polarization instability (pseudoergodicity) of nonergodic relaxor.

The evolution of piezoelectric response has also been added in Fig. S1, a few more discussions have been added in text in red (Page 4).

Fig. R1. Evolution of d_{33} with changing composition.

4. Interpretation about PNRs from the bright-field image in Figure S2 is not clear. How the ellipsoid shape has been obtained for the PNRs? Figure S2, shows different contrast across the image, what is the reason for the differences in the top and bottom regions of the image? How much diffraction contrast from sample mistilt influence the domain interpretation from bright-field images.

Reply: The shape of PNRs was described as ellipsoid shape in previous work from TEM images. There is a little difference for the PNRs shape in this work, so we have revisited about this description and reedited the corresponding discussion in text in red (Page 8). Because the TEM sample was prepared by ion milling, the thickness of sample decreases with approaching to the hole, leading to the contrast differences in the top and bottom regions of the image. Yet the contrast difference of the wedge shape sample does not influence the contrast caused by the PNRs. In order to show the features of PNRs more clearly, HADDF STEM and the corresponding polarization mapping has been added in the revised text .

5. A high-resolution STEM imaging can be helpful here to understand the polarization changes and determine the domain shapes instead of just bright-field image.

Reply: Yes, thanks for your good suggestion, HADDF STEM has been measured and added in the revised text (Fig. 3).

6. Figure 3e label should be -40 kV rather than 0V?

Reply: Yes, thanks for your careful review, we have redrawn the corresponding figure.

7. Which diffraction spots have been used to obtain the dark field images in Figure 3 a2 and c2?

Reply: The corresponding diffraction spot used for dark field images has been marked in the corresponding figure and illustrated in the figure caption.

8. From the in situ high-energy synchrotron X-ray diffraction study, pseudocubic phase has been identified at the zero electric field which transforms to monoclinic phase at higher field. The phase diagram in Figure S1a revealed the composition used in this study are from the boundary between rhombohedral and tetragonal phases. It is unclear if there is any signature of rhombohedral or tetragonal phases in the diffraction data at zero field sample.

Reply: Yes, the compositions chosen in this work are located at rhombohedral-tetragonal phase boundary, this can be further confirmed by the polarization mapping results using HADDF STEM. Owing to the resolution ratio of XRD, the symmetry of PNRs cannot be distinguished. Instead, a pseudocubic average phase structure would be found for relaxor ferroelectrics, this phenomenon has been widely reported in various relaxor materials. (100), (110), (111), (200), (220), (222) and (400) diffraction peaks are all single peak without observable split at zero field. Other space groups such R3c, R3m and P4mm were also used for Rietveld refinement, showing obvious larger R_{wp} than Pm-3m. Therefore, the studied sample exhibits rhombohedral-tetragonal coexistence at local scale but pseudocubic for average structure. A few more discussions about this result have been added in text in red (Page 10).

9. The fitting of P-E loop in Figure 4d is unclear. Please explain the meaning of the fitted P-E loop

in the figure.

Reply: Yes, in order to explain the evolution of P - E shape, especial for the double P - E loop around T_c , of BaTiO₃ with changing temperature, the phenomenological theory of thermodynamics was proposed to fit the P - E loop using the Landau-Ginzburg-Devonshire theory [Phys. Rev. 91, 513, 1953]. Strictly speaking, the parameters for fitting should be gotten from single crystals or theory calculation to quantitatively analyze the free energy profile of a ferroelectric single crystal. In this work, the free energy profile of a ceramic with special double-like P - E is fitted based on the reference of previous research results and parameters. Even though it might be not precise enough for quantitative analysis, it can be used for the qualitative analysis about the relative free energy state between relaxor and normal ferroelectric phases. Some discussions have been added in text in red (Page 13).

10.Explain where the findings in this study stand with previous phase field modeling work by Li, F. et al. The origin of ultrahigh piezoelectricity in relaxor-ferroelectric solid solution crystals. Nat. Commun. 7, 13807.

Reply: Thanks for the good suggestion. A typical work about high piezoelectricity using phase field modeling has been done by Li, F. et al. Some discussions about this reference have been added in text in red (Page 13).

Reviewer #2 (Remarks to the Author):

The paper reports results of electric, TEM, and XRD measurements of a nonergodic relaxor (Pb_{0.917}La_{0.083})(Zr_{0.65}Ti_{0.35})_{0.97925}O₃ ceramic sample to demonstrate that the electric field induced relaxor-normal ferroelectric phase transition is the origin of its large electrostrain. The double-like P - E loop and changes of TEM images and XRD patterns under electric field clearly show the relaxor-normal ferroelectric phase transition. The pseudo-ergodicity produced by the field induced phase transition in the nonergodic relaxor is informative to understand electromechanical responses of relaxor ferroelectrics. **However, the field induced relaxor-normal ferroelectric phase transition is not surprising and therefore lacks the novelty for publication in Nature Communication.** In addition, the following points should be addressed.

Reply: Yes, as the reviewer mentioned, relaxor-normal ferroelectric phase transition has been widely reported in the previous researches. However, the novelty of this work is not the founding of relaxor-normal ferroelectric phase transition but the new insight into the process of relaxor-normal ferroelectric phase transition as well as the corresponding mechanism of large electromechanical response. The achievement of large electrostrain was attributed to the ergodicity in previous works, which causes spontaneous backward switching from high-field ferroelectric to ergodic relaxor state. In this work, we have proved that the large electrostrain is achieved in nonergodic relaxor phase, which exhibit pseudoergodicity under periodicity electric field and can also be triggered from normal ferroelectric state by electric field.

T_f in the V-F law of eq. (1) is used without its definition in the introduction. The definition of T_f should be added in the introduction.

Reply: Yes, the definition of T_f has been added in the introduction in red (Page 3). Thanks for your good suggestion.

T in eq. (1) does not match with T_m in the inset of Fig. 1. It should be replaced by T_m .

Reply: Yes, eq. (1) has been reedited according to the suggestion. Thanks.

E in the line 122 does not match with E_b in eq. (1). It should be replaced by E_b .

Reply: Yes, the corresponding error has been corrected. Thanks.

The authors analyzed the XRD patterns under electric fields using Rietveld refinements. However, powder diffraction data under an electric field cannot be usually analyzed by single phase Rietveld refinements. No refined parameters and reliable factors of the analyses are shown in the manuscript. The authors should give the detail of the methods and results of the Rietveld refinements.

Reply: Yes, the detail of the refinement results has been added as Table S1 and some discussions have been added in text in red (Page 10). According to the reliability factor of weighted patterns and the goodness of fit indicator, the Rietveld refinement results should be reliable. Thanks again for the suggestion.

Reviewer #3 (Remarks to the Author):

The manuscript presents extensive structural, microstructural, electrical and electro-mechanical property data from a PLZT relaxor composition. The main claim of the manuscript, as far as I understand, is that the Authors report a material that is nominally in the non-ergodic relaxor state according to electrical characterisation, however it experiences a reversible electric field induced strain mechanism that is not consistent with past observations in this material state. Overall, I would say the data are reasonable and likely support the conclusions. However, I would also suggest significant improvement could be made in the manuscripts clarity by at least addressing the queries below.

I don't clearly see the differentiation between the strain mechanism shown here and those in other systems. Electric field induced lattice strains (without clear phase transitions) are observed in compositions of NBT-BT-KNN and NBT-ST for example. Applying a similar analysis (of the 45 degree azimuth from HEXRD) in those cases would likely yield a similar result. Can the Authors be more clear about what is unique here?

Reply: Yes, many researches, including our group's studies,^[38] have been done to quantitatively analyze the strain contribution using in-situ HEXRD.^[39-41] In order to show the contributions to the excellent electromechanical properties of the studied PLZT system more clearly, quantitative decomposition was further taken and has been added in Figs. 5d-e, S5 and S6. A few more discussions have also been added in text in red (Page 11). Different from BNT-based ceramics, the studied PLZT ceramics exhibits a relatively large lattice strain contribution, thus leading to high piezoelectric response and low-field driven large strain behavior.

The Authors use the description "canonical relaxors" with no explanation of their meaning of this description. Please ensure its provided on first use of the word.

Reply: Yes, the description of "canonical relaxors" has been changed to be "typical relaxors".
Thanks for your suggestion.

The Authors reference the "...nonergodic and ergodic relaxor states coexist in a wider temperature

range...” at the bottom of page 3. However, this doesn't seem to match their explanation of what these states are in the previous sections. How do the Authors reconcile that result with the description of these relaxor states they've provided? Perhaps this is the first sentence of the first paragraph on page 4 where they state, “It seems that the ergodicity can be influenced by the application of external electric field.”, but please expand on this to explain what is meant?

Reply: Yes, a few more discussions about this question have been added in text in red (Page 4). In order to explain the unusual reversible phase transition in nonergodic relaxor zone, coexistence of nonergodic and ergodic relaxor states were proposed in some previous works. Yet nonergodic-ergodic coexisted phase cannot well explain the critical phenomenon found in the studied sample. Therefore, we proposed a new insight of field induced pseudoergodicity to explain the critical phenomenon close to nonergodic-ergodic boundary.

Page 5 the Authors refer to Figure 1b. However the figure has no b). I suspect this is just referring to the inset.

Reply: Yes, the corresponding error has been corrected (Page 5). Thanks for your careful review.

The Authors state several time that the response is not consistent with polarisation or converse piezoelectric effect, however then go on to use the words such as “...new insights into the polarization reorientation process...”. What does reorientation mean if not rotation or intrinsic converse behaviour, this is not at all clear. At the very least a schematic diagram of what the Authors suggest would be required, how are they differentiating this mechanisms from others?

Reply: Yes, thanks for your good suggestion. In piezoelectrics, polarization rotation (polarization angle change) and converse piezoelectric effect (polarization magnitude change) can be found under electric field. For the studied relaxor ferroelectric, polarization reorientation occurs as normal-relaxor-normal ferroelectric ($P1 \rightarrow 0 \rightarrow -P2$) during applying reversed electric field. The corresponding discussion has been rewritten and a schematic diagram has been added in Fig. 6d to show this process in detail. At the same time, the quantification of strain contributions was added in Figs. 5, S5 and S6 to support the conclusion. Some discussions have been added in text in red (Pages 11-13).

Presumably looking at the XRD data in sector 0 of Fig S5 would provide some insight as to the field induced process. These data are shown in Figure S4, but without any analysis on peak intensities or widths with field. A plot of each of the peak widths vs field would provide some information about particular distortions occurring, if they indeed are.

Reply: Yes, that is a good suggestion. Accompany the phase transition and domain texture process, obvious change of peak position and peak width can be detected. In order to show these contributions, further analysis about the XRD data has been added in text in red (Page 11, Figs. S5, S6 and 5). Thanks.

The analysis of the 45 degree XRD sector to conduct phase analysis is a little odd. Particularly in this state where each grain orientation must be undergoing separate and unique distortions to the applied field, the 45 degree sector will inevitably look like a low symmetry structure due to the convolution of these effects. Saying it is monoclinic here is possible misleading. The authors should clearly state that each grain orientation set is undergoing unique distortions and applying a single symmetry is not ideal, but perhaps is the best option without employing a more complete analysis method such as using the strain orientation distribution functions as done by other Authors.

Reply: Yes, thanks for your professional advice. Some discussions about this comment have been added in text in red (Page 10-11). Moreover, further analysis about the strain orientation distribution functions was also added in the revised text.

The Authors should not use non-scientific language such as “marvelous” to describe the materials behaviour/state.

Reply: The corresponding description has been revised in text in red (Pages 1 and 6). Thanks.

Related, the English quality of the paper could be improved significantly. There are many grammatical and typographical errors throughout. A quality technical English language edit would be highly advised before resubmitting.

Reply: Yes, we have improved the English quality carefully. We are willing to accept further optimization by the language edit of this journal if necessary.

Thank you very much for your consideration of the manuscript and please feel free to contact us if any additional information is required.

Best wishes

Jun Chen

REVIEWER COMMENTS

Reviewer #1 (Remarks to the Author):

I find authors have made a good effort in modifying the manuscript based on the reviewers' comments. But I still find some concerns over the interpretations of some of the data, especially microscopy images. In the response of my question 4, authors mentioned as ellipsoidal shape is observed in other relaxor systems, in PLZT, we should assume ellipsoidal shape, I am still not sure what is the basis of this assumption. Also, as the sample is prepared by mechanical wedge polishing, the image contrast in bright-field TEM images will always have the influence of sample thickness and mistilt which can be clearly seen in Figure 3 a (difference between top and bottom). Authors should clearly mention these to be potential sources of error in finding the size of these PNRs. I would thank the authors to provide high-resolution HAADF STEM image. But I am not sure how they found polarization using this image. I would guess this would be either A or B sub lattice displacements. I would suggest mentioning some details either in main text or in the method section. I am again confused how they made these phase boundaries of R, T and PC phases. In the current form, it seems to be drawn arbitrarily. I also found it interesting, from Fig 3a bright-field TEM image, authors found PNRs of 10-25 nm of width but from phase boundaries (domains) drawn in Figure 3b, PNRs width can be found between roughly 2-8 nm. I would suggest authors to clarify this in the manuscript.

Authors mentioned "the slush-like PNRs in pseudocubic (PC) matrix". In Figure 3b, authors denote minimal polarization or non-polar regions as for PC phase. It is not clear to me why authors mention PC as matrix here. In the recently proposed slush model for relaxors, there exists no non-polar matrix which contrasts from the older model for relaxors where PNRs exist in a non-polar matrix. The slush model of relaxors proposed existence of multiple phases with low angle domain walls. Minimal or no polarization can be found using STEM images due to projection effects as well, in the case when polarization points out of the plane. I would suggest authors to clarify about phase determination in the manuscript.

I would suggest significant changes in the current form of the manuscript based on the above comments before its publication.

Reviewer #2 (Remarks to the Author):

Results of Rietveld refinements were added in Table S1 to address my comments. However, refined atomic coordinates and atomic displacement parameters are not shown. These structure parameters are the most important in Rietveld structure refinements. I recommend that the authors add these refined structure parameters in Table S1.

Dear Editor:

Thank you very much for giving us an opportunity to revise our manuscript entitled “**Giant dynamic electromechanical response via field driven pseudo-ergodicity in nonergodic relaxors (Ms. No. NCOMMS-22-44609A)**”. We also thank the reviewers for their constructive and valuable comments. We have addressed the comments raised by the reviewers, and the revisions were highlighted in red in the revised manuscript. Point by point responses to the reviewers’ comments are also listed below.

Reviewer #1 (Remarks to the Author):

I find authors have made a good effort in modifying the manuscript based on the reviewers’ comments. But I still find some concerns over the interpretations of some of the data, especially microscopy images. In the response of my question 4, authors mentioned as ellipsoidal shape is observed in other relaxor systems, in PLZT, we should assume ellipsoidal shape, I am still not sure what is the basis of this assumption. Also, as the sample is prepared by mechanical wedge polishing, the image contrast in bright-field TEM images will always have the influence of sample thickness and mistilt which can be clearly seen in Figure 3 a (difference between top and bottom). Authors should clearly mention these to be potential sources of error in finding the size of these PNRs. I would thank the authors to provide high-resolution HAADF STEM image. But I am not sure how they found polarization using this image. I would guess this would be either A or B sub lattice displacements. I would suggest mentioning some details either in main text or in the method section. I am again confused how they made these phase boundaries of R, T and PC phases. In the current form, it seems to be drawn arbitrarily. I also found it interesting, from Fig 3a bright-field TEM image, authors found PNRs of 10-25 nm of width but from phase boundaries (domains) drawn in Figure 3b, PNRs width can be found between roughly 2-8 nm. I would suggest authors to clarify this in the manuscript.

Reply: Thanks for your good comments about the TEM and HAADF STEM results.

1. About the shape of PNRs, indeed, the sample thickness would affect the image contrast. We reprepared another sample with higher quality to eliminate the influence from the difference of thickness, as shown in Fig. 3a and S3. As the reviewer suggested, the shape of PNRs should be described based on the results found in this work instead of assumption from other relaxor systems.

We have reedited the corresponding descriptions in the manuscript in red (Page 8).

2. The polarization displacement vectors were not drawn arbitrarily in Fig. 3, they were drawn based on the quantitative analysis of the position of each atom. The process is shown schematically in the following figure. This method has also been widely used in previous studies, and the detail mechanism can be found in these references: Micron 2018, 113, 99–104; Microscopy and Microanalysis 2022, 0, 1–9. Details for the calculating process have been added in the experimental part (Page 17). Thanks.

Fig. R1. Flow diagram for the process of calculating polarization displacement according to HAADF STEM.

3. Yes, obvious difference of the PNR size can be seen between bright field TEM and HAADF STEM images. This should be mainly ascribed to the limited observation area of HAADF STEM

image. If a larger region was taken for HAADF STEM, it is inaccurate for calculating polarization vectors owing to the relatively low resolution ratio of each atom column. HAADF STEM images were taken at different regions, it is hard to reveal the feature of phase coexistence for the images taken inside a single PNR (as shown in Fig. 3b). Therefore, an image taken at around the domain boundary or at a region with small PNRs was chosen at the same time (Fig. 3c). It includes the local symmetry feature of this sample, but the size of PNRs cannot represent the characteristics of PNRs. We have reedited to corresponding description in the manuscript in red (Page 8). Thanks for your professional comments.

Authors mentioned “the slush-like PNRs in pseudocubic (PC) matrix”. In Figure 3b, authors denote minimal polarization or non-polar regions as for PC phase. It is not clear to me why authors mention PC as matrix here. In the recently proposed slush model for relaxors, there exists no non-polar matrix which contrasts from the older model for relaxors where PNRs exist in a non-polar matrix. The slush model of relaxors proposed existence of multiple phases with low angle domain walls. Minimal or no polarization can be found using STEM images due to projection effects as well, in the case when polarization points out of the plane. I would suggest authors to clarify about phase determination in the manuscript.

Reply: Yes, whether there is nonpolar matrix is a big debate in previous researches, and the out-of-plane polarization would reveal as nonpolar state in STEM images. According to the statistics of polarization vectors in 15 different regions, the proportion of nonpolar vectors is quite small. Therefore, these nonpolar regions should be mainly related to out-of-plane domains, and the slush-like PNRs model without nonpolar matrix should be more suitable for the results found in this work. The corresponding discussion has been added in the manuscript in red (Page 8 and Fig. 3d).

I would suggest significant changes in the current form of the manuscript based on the above comments before its publication.

Reply: Yes, we have optimized the corresponding discussion according to the reviewer’s comments. Thanks for your professional suggestions.

Reviewer #2 (Remarks to the Author):

Results of Rietveld refinements were added in Table S1 to address my comments. However, refined atomic coordinates and atomic displacement parameters are not shown. These structure parameters are the most important in Rietveld structure refinements. I recommend that the authors add these refined structure parameters in Table S1.

Reply: Yes, we have added the other structure parameters of Rietveld refinements in Table S1.

Thanks for your good suggestions.

Thank you very much for your consideration of the manuscript and please feel free to contact us if any additional information is required.

Best wishes

Jun Chen

REVIEWERS' COMMENTS

Reviewer #1 (Remarks to the Author):

I find authors have done a great improvement in the latest draft in terms of microscopy data and analysis based on the reviewers' comments. Authors have also explained their method of extracting polarization maps using HAADF STEM images. But I still find few information missing in their Methods section for TEM experiments, what method has been used to correct drift and scan distortions from HAADF STEM images before extracting atomic displacements from them. Also, Figure 3b&c show polarization or net displacement maps which I would assume authors have extracted as net displacements of Pb/La sublattice with respect to their nearest Zr/Ti sublattice. Please mention the semi-convergence angle and collection angles used. I would suggest authors to clearly write experimental information in the Methods section such that their microscopy results can be easily reproduced. I would also suggest using some more transparency in the atomic resolved HAADF STEM images in Figure 3b&c on which displacement maps are overlaid such that arrow size and color can be visualized better for the readers as it is hard to see these displacement vectors in the current form. I recommend that the manuscript can be published after the minor revisions.

Dear Editor:

Thank you very much for giving us an opportunity to revise our manuscript entitled “**Giant dynamic electromechanical response via field driven pseudo-ergodicity in nonergodic relaxors (Ms. No. NCOMMS-22-44609B)**”. We also thank the reviewers for their constructive and valuable comments. We have addressed the comments raised by the reviewers, and the revisions were highlighted in red in the revised manuscript. Point by point responses to the reviewers’ comments are also listed below.

Reviewer #1 (Remarks to the Author):

I find authors have done a great improvement in the latest draft in terms of microscopy data and analysis based on the reviewers’ comments. Authors have also explained their method of extracting polarization maps using HAADF STEM images. But I still find few information missing in their Methods section for TEM experiments, what method has been used to correct drift and scan distortions from HAADF STEM images before extracting atomic displacements from them. Also, Figure 3b&c show polarization or net displacement maps which I would assume authors have extracted as net displacements of Pb/La sublattice with respect to their nearest Zr/Ti sublattice. Please mention the semi-convergence angle and collection angles used. I would suggest authors to clearly write experimental information in the Methods section such that their microscopy results can be easily reproduced. I would also suggest using some more transparency in the atomic resolved HAADF STEM images in Figure 3b&c on which displacement maps are overlaid such that arrow size and color can be visualized better for the readers as it is hard to see these displacement vectors in the current form. I recommend that the manuscript can be published after the minor revisions.

Reply: Thanks for your positive comments. The detailed experimental methods about the HAADF STEM results have been added in the experimental section, and Figure 3b&c has been redrawn in order to show the polarization mapping more clearly. Thanks again.

Thank you very much for your consideration of the manuscript and please feel free to contact us if any additional information is required.

Best wishes

Jun Chen